# Aminopeptidase MNP-1 triggers intestine protease production by activating *daf-16* nuclear location to degrade pore-forming toxins in *Caenorhabditis elegans*

**Feng Chen[1], Cuiyun Pang[1], Ziqiang Zheng[1], Wei Zhou[1], Zhiqing Guo[1], Danyang Xiao[1], Hongwen Du[1], Alejandra Bravo[2], Mario Soberón[2], Ming Sun[1], Donghai Peng[1]***

**1** State Key Laboratory of Agricultural Microbiology, Huazhong Agricultural University, Wuhan, Hubei, People's Republic of China, **2** Instituto de Biotecnología, Universidad Nacional Autónoma de México, Cuernavaca, Morelos, Mexico

* donghaipeng@mail.hzau.edu.cn

**Data Availability Statement:** The numerical data used in all figures are included in S1 Data. The microarray data are available on the NCBI Gene

## Abstract

Pore-forming toxins (PFTs) are effective tools for pathogens infection. By disrupting epithelial barriers and killing immune cells, PFTs promotes the colonization and reproduction of pathogenic microorganisms in their host. In turn, the host triggers defense responses, such as endocytosis, exocytosis, or autophagy. *Bacillus thuringiensis* (Bt) bacteria produce PFT, known as crystal proteins (Cry) which damage the intestinal cells of insects or nematodes, eventually killing them. In insects, aminopeptidase N (APN) has been shown to act as an important receptor for Cry toxins. Here, using the nematode *Caenorhabditis elegans* as model, an extensive screening of *APN* gene family was performed to analyze the potential role of these proteins in the mode of action of Cry5Ba against the nematode. We found that one APN, MNP-1, participate in the toxin defense response, since the *mnp-1(ok2434)* mutant showed a Cry5Ba hypersensitive phenotype. Gene expression analysis in *mnp-1 (ok2434)* mutant revealed the involvement of two protease genes, *F19C6.4* and *R03G8.6*, that participate in Cry5Ba degradation. Finally, analysis of the transduction pathway involved in *F19C6.4* and *R03G8.6* expression revealed that upon Cry5Ba exposure, the worms up regulated both protease genes through the activation of the FOXO transcription factor DAF-16, which was translocated into the nucleus. The nuclear location of DAF-16 was found to be dependent on *mnp-1* under Cry5Ba treatment. Our work provides evidence of new host responses against PFTs produced by an enteric pathogenic bacterium, resulting in activation of host intestinal proteases that degrade the PFT in the intestine.

## Author summary

Pore-forming toxins (PFTs) have a crucial role in the pathogenesis of microbial pathogens. Meanwhile, hosts mediate different strategies to resist PFTs at the cellular level, such as endocytosis, exocytosis, autophagy, and MAPK or inflammasome signaling pathways.

Expression Omnibus(GEO)(https://www.ncbi.nlm.nih.gov/geo/), under the record GSE163072 (https://www.ncbi.nlm.nih.gov/geo/query/acc.cgi?acc=GSE163072).

**Funding:** This study was supported by the National Natural Science Foundation of China (U20A2040 to M.S., 31970076 to M.S., 31970075 to D.P., and 31770116 to D.P.), National Key R&D Program of China (2017YFD0200400 to D.P.). The funders had no role in study design, data collection and analysis, decision to publish, or preparation of the manuscript.

**Competing interests:** The authors have declared that no competing interests exist.

Here we show that the nematode *Caenorhabditis elegans* respond to PFTs inducing their proteolysis by up regulating intestinal proteases through the induction of FOXO transcription factor DAF-16 that showed nuclear location. Our results indicate that animals not only trigger defense mechanisms after PFTs intoxication at the cellular level, but also could eliminate PFTs in the intestine lumen, an effective additional strategy for animals to resist PFTs, especially those produced by enteric bacterial pathogens. Since, the FOXO transcription factor DAF-16 is conserved from nematodes to mammals, it is possible that the degradation of PFT by modulating intestinal proteases may be a common defense strategy against bacterial infections.

## Introduction

The pore-forming toxins (PFTs) play important roles in bacterial pathogenesis, including many important human pathogens, such as *Staphylococcus aureus*, *Streptococcus pyogenes*, *Vibrio cholera*, *Clostridium perfringens*, *Enterococcus faecalis*, among others [1–6].The different PFTs bind to membrane receptors such as membrane proteins, lipids, or cholesterol triggering conformational changes in these toxins leading to oligomer formation, and membrane insertion, resulting in their pore formation activity on the cell membrane. It has been documented that even at sublethal concentrations, the action of PFTs would modify the physiological state of cells by activating different cellular responses. Host cells could remove PFTs on plasma membrane by endocytosis and exocytosis, and may induce degradation of PFTs by activating autophagy. In addition, enhance resistance toward PFTs is also carried out by MAPKs pathways or by the inflammasome complex, or by triggering cell death programs such as apoptosis, necrosis or pyroptosis [7–11]. For example, some PFTs could be removed by membrane repair mechanisms such as endocytosis and exocytosis that are the most accepted membrane repair mechanisms [12]. In the case of cholesterol-dependent cytolysins (CDCs), pore formation induces an intracellular $[Ca^{2+}]$ increase in the intoxicated cells, promoting internalization of the PFT in caveolaes, in a $Ca^{2+}$ dependent endocytosis process, that finally leads to PFTs degradation in lysosomes [13, 14]. Likewise, exocytosis of α-toxin, CDCs or Cry5Ba toxins was shown to be dependent on the rise of intracellular $[Ca^{2+}]$ induced by the PFTs, which in turn enhances shedding of microvesicles on the membrane that contains the PFTs pores [15–17]. In addition, different PFT activate signal transduction cascades like MAPKs and autophagy to remove PFTs from the membrane, and resist cell damage [18]. For example, the decrease in intracellular potassium induced by some PFTs, could activate the p38/MAPK pathway, which up-regulate multiple downstream genes involved in the activation of unfolded protein response (UPR) and innate immunity against PFTs intoxication [18, 19]. The MAPK cascade is also known to regulate the phosphorylation level of the nuclear receptor fushi tarazu factor 1 (FTZ-F1), and the phosphorylation status of FTZ-F1 directly impacts the expression of the Cry1Ac receptor in *Plutella xylostella*, ultimately determining the tolerance or sensitivity of *P. xylostella* to Cry1Ac[20, 21]. Moreover, autophagy has been documented as another efficient way to protect host cells against different PFTs, such as *V. cholerae* cytolysin (VCC), α-Haemolysin (HlyA) and α-toxin [22–24]. In the case of *Caenorhabditis elegans*, survival of this nematode to Cry5Ba PFT depend on autophagy activation that participates in degradation of the toxin pores [25].

*Bacillus thuringiensis* (Bt) is a spore-forming bacterium that upon sporulation, produces parasporal crystal proteins (Cry) that have specific toxicity to different insects, nematodes, mites, or protozoa species [26]. These Cry toxins disrupt the host intestine after the target

organism ingest the parasporal crystal, leading to the host death. The Bt spores germinate and reproduce in the host cadaver [27, 28]. Bt insecticidal proteins that accumulated in the crystal inclusions are PFTs which are the major Bt virulence factors. Based on their structural features, they can be divided into several families, such as the Cry three-domain toxins, the α-PFTs toxins now named App and β-PFTs such as aegerolysins now named as Gpp, etc [29]. The three-domain Cry proteins have been proposed to generate pores on the insect larvae or nematodes intestinal cell membrane. Cry proteins bind to receptors (such as aminopeptidase N, alkaline phosphatase or cadherin among others) and undergo conformation changes forming oligomers, that ultimately insert into the membrane to generate pores on the plasma membrane of midgut cells [30].

*C. elegans* is a powerful model system to study host responses to PFTs. Cry5Ba is a nematocidal Cry toxin, that after binding to a glycolipid receptor, disrupt the nematode intestine cells leading to worm death [31–33]. It has been shown that worms resist Cry5Ba intoxication by inducing multiple conserved innate immune responses, including the activation of p38 and c-Jun/MAPKs, the endoplasmic reticulum UPR and hypoxia response pathways, as well as the DAF-2 insulin/insulin-like growth factor-1 signaling pathway [34–38]. Furthermore, it was reported that at the intracellular level, worms could initiate endocytosis, exocytosis, and autophagy as defense strategies from Cry5Ba toxin action. However, how worms reduce toxicity when exposed to lethal concentration of Cry5Ba inside the intestine lumen has not been clearly elucidated.

In this study, we found that *C. elegans* modulate the transcription factor DAF-16 nuclear location to promote protease expression in the gut lumen, which in turn degrade Cry5Ba toxin. Our work explains how animals clear and inactivate PFTs in the intestine lumen, which represents a novel and additional strategy activated in the hosts to defend against pathogenic bacteria.

## Results

### 1. MNP-1 is required for protecting *C. elegans* against Cry5Ba toxin

The identification of *C. elegans* receptors involved in Cry5Ba mode of action revealed a role of apical cell glycolipids and cadherin CDH-8 [31, 39]. In the case of insects, cadherin and aminopeptidase N (APNs), among other proteins, act as Cry toxin receptors [40, 41]. Therefore, we analyzed whether APNs may also function as receptors for Cry5Ba in *C. elegans*. The APN protein family are enzymes that cleave neutral amino acids from the N terminus of polypeptides, which serve a variety of functions in a wide range of species from nematodes to mammals [42]. Two reported APN receptors of Cry proteins reported in insects contain two conserved domains, the catalytic center domain M1_APN_2 and ERAP1-C domain [43]. Thus, we identified all APNs that have the M1_APN_1 like domain in *C. elegans* genome. We found eight APNs containing both the catalytic center domain M1_APN_1 like and ERAP1-C domain, three APNs that only have the M1_APN_1 like domains, and other three APNs that have the Peptidase_M1_N domains (S1 Table, S1A–S1C Fig). To analyze the potential role of these APN´s as Cry5Ba receptors, we analyzed the toxicity of Cry5Ba toxin to APN mutants, since receptor mutations have been shown to be linked to resistance to different Cry toxins in several organisms [44, 45]. A total of eight APNs mutants were obtained from *Caenorhabditis* Genetics Center (CGC) and evaluated by comparing their susceptibility to Cry5Ba with that of the wild-type N2 worms (S2 Table). The expression of other six APNs was down regulated by RNA interference (RNAi) assays to compare the susceptibility of these silenced animals after RNAi to Cry5Ba toxin in relation to the control nematodes. In these RNAi assays the animals were feed with HT115 *E. coli* cells that were transformed to express the different dsRNA, and

the dsRNA expression was induced with IPTG as reported [46]. Control animals were treated with non-induced transformed *E.coli* cells. None of the *apn* mutants nor *apn*-silenced worms by RNAi that were analyzed showed a significant resistance phenotype to Cry5Ba, compared with the control animals, suggesting that the 14 APNs analyzed do not participate as receptors for Cry5Ba toxin in *C. elegans* (S2 Table).

However, we found out that one of the APN mutants, the *mnp-1(ok2434)*, was 8.78-fold more sensitive to Cry5Ba (medium lethal concentration [$LC_{50}$] value of 1.929 ± 0.28 μg/mL) than N2 animals ($LC_{50}$ value of 16.926 ±0.8 μg/mL) (Fig 1A, and Table 1), suggesting that MNP-1 may be involved in protecting *C. elegans* against Cry5Ba toxin. To confirm the defense role of MNP-1 against Cry5Ba in *C. elegans*, we first analyzed *mnp-1* expression in the worms. We generated a transgenic *mnp-1p::gfp* worms that express GFP protein fused to MNP-1 protein and confirmed that it is expressed in all stages from the egg to adult (S2 Fig). Secondly, the susceptibility to Cry5Ba was analyzed by using a growth inhibitory (GI) assays where the *mnp-1(ok2434)* mutant worms showed increased susceptibility to Cry5Ba than N2 control animals (Fig 1B). Moreover, lifespan measurements were also performed in N2 and *mnp-1(ok2434)* mutant worms upon exposure to Cry5Ba and the control bacterial strain OP50 (Fig 1C, Table 1). These assays showed that *mnp-1(ok2434)* mutant worms showed a significantly 2.6 fold lower lifespan than N2 control worms under Cry5Ba treatment. It is important to notice that the lifespan of N2 and *mnp-1(ok2434)* showed no difference under OP50 treatment. (Fig 1C, Table 1). Images of the mutant worms confirmed that in the presence of the Cry5Ba, the *mnp-1(ok2434)* mutant worms were more severely intoxicated compared to wild-type N2 worms, as evidenced by their smaller body-size (Fig 1D). These results confirm that *mnp-1 (ok2434)* mutant is more sensitive to Cry5Ba than wild-type N2 worms.

To confirm the importance of MNP-1 in a defense response to Cry5Ba toxicity. The expression of MNP-1 was reduced in the wild type N2 worms by using RNAi (S3 Fig). Fig 1E shows that MNP-1-silenced individuals were more sensitive to Cry5Ba toxin, supporting that the Cry5Ba sensitive phenotype is caused by reduction of MNP-1 expression (Fig 1E, Table 1). Finally, we confirmed the role of MNP-1 by complementing the *mnp-1(ok2434)* mutant with the wild type *mnp-1* gene. The *mnp-1(ok2434)* mutant worms were transformed with the *mnp-1* gene under regulation of its native promotor, showing that the rescued worms had similar sensitivity to Cry5Ba toxin when compared to the wild type N2 worms (Fig 1D and 1F), indicating that the *mnp-1* gene was sufficient to restore wild type susceptibility to Cry5Ba toxin in the *mnp-1(ok2434)* mutant. Taken together, these data demonstrate that MNP-1 is required for *C. elegans* defense against Cry5Ba toxin.

## 2. MNP-1 is specifically involved in defense to Cry5Ba and Cry21Aa but not to other nematicidal toxins such as App6Aa PFT

There are several classes of nematicidal toxins produced by Bt such as Cry5Ba, Cry21Aa and App6Aa (before named Cry6Aa). The Cry5Ba and Cry21Aa belong to the Cry three domain family, while App6Aa toxin belongs to the ClyA-type α-pore-forming toxin family [29, 47, 48]. To test whether MNP-1 is required for protecting *C. elegans* against other nematicidal PFTs, the susceptibility of *mnp-1(ok2434)* mutant to App6Aa and Cry21Aa was compared with the wild-type N2 animals. As in the case of Cry5Ba, both the mortality (Fig 2A) and the GI assays (Fig 2B) showed that *mnp-1(ok2434)* mutated animals were more sensitive to Cry21Aa than wild type N2 animals (Fig 2A and 2B, Table 1). However, in the case of App6Aa, the *mnp-1 (ok2434)* mutant and wild type N2 animals showed similar sensitivity when exposed to App6Aa toxin (Fig 2C and 2D, Table 1).

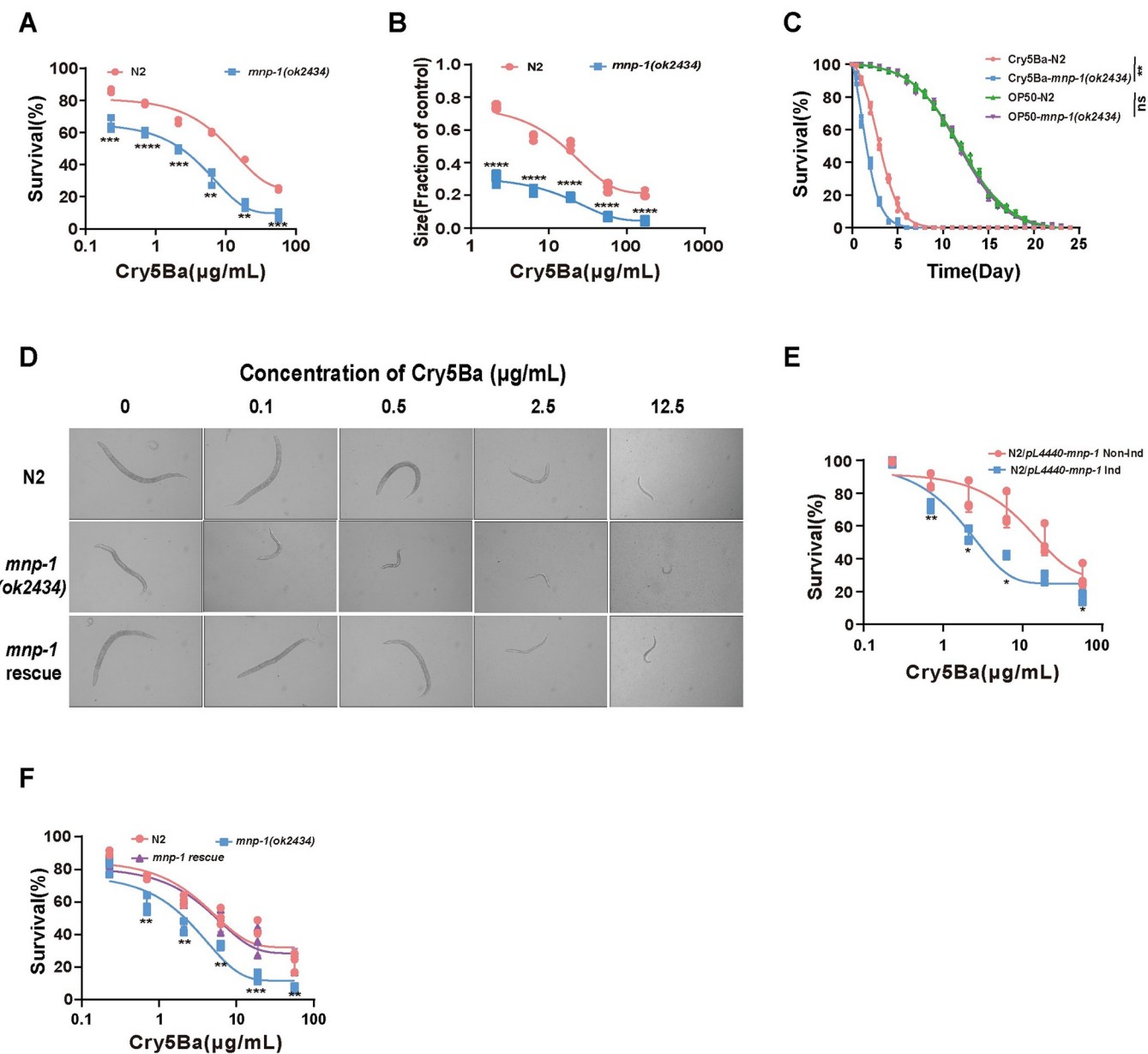

**Fig 1. MNP-1 is required for protecting *C. elegans* against Cry5Ba toxin.** (A), a dose-dependent mortality assay comparing sensitivity of *mnp-1(ok2434)* mutant and wild-type N2 worms exposed to purified Cry5Ba toxin. *N* = 3 independent experiments containing at least 30 worms. (B), a dose-dependent growth assay comparing sensitivity of *mnp-1(ok2434)* mutant and wild-type N2 worms exposed to purified Cry5Ba. *N* = 3 independent experiments containing at least 30 worms. (C), the lifespan analysis of wild type N2 and *mnp-1(ok2434)* mutant exposed to Cry5Ba and OP50. *N* = 3 independent experiments containing at least 100 worms. Data were analyzed by using Kaplan–Meier followed by log-rank Mantel-Cox tests. (D), representative images of the worms in each treatment as follows: L1 larvae of N2, *mnp-1(ok2434)* mutant, and *mnp-1(ok2434)* rescued with *mnp-1* gene and observed under a light microscope. *N* = 3 independent experiments. (E), a dose-dependent mortality assay comparing sensitivity of *mnp-1* RNAi and control worms exposed to Cry5Ba. N2 worms were fed with *mnp-1* dsRNA fragment to silence *mnp-1* gene (N2/*pL4440-mnp-1* Ind) expression. N2 worms fed on uninduced *E. coli* HT115 transformed with the vector *pL4440-mnp-1*(N2/*pL4440-mnp-1* Non-Ind) were used as controls. The RNAi worms were exposed to purified Cry5Ba after reaching L4 stage. *N* = 3 independent experiments containing at least 30 worms. (F), a dose-dependent mortality assay of N2, *mnp-1(ok2434)* mutant and *mnp-1* rescued worms after treatment with Cry5Ba. At least 30 worms were used in each repeat. *N* = 3 independent experiments. Data points represent the mean values of three independent replicates, error bars denote the SD in A, B, C, E and F. The *p*-value was determined by Unpaired *t*-test (Welch's correction for unequal variances), ****$p < 0.00001$, ***$p < 0.001$, **$p < 0.01$, *$p < 0.05$ showing significant differences.

**Table 1. Data analysis of the LC$_{50}$ values of Cry5Ba, App6Aa and Cry21Aa against worms and lifespan assay.**

**Cry5Bay**

| Strain | LC$_{50}$(μg/ml)[a] | 95% confidence interval (μg/ml) | Standard Deviation | *p* value relative to N2 | Relative sensitivity[b] |
|---|---|---|---|---|---|
| Wild type (N2) | 16.926 | 12.436–23.7 | 0.800 | / | |
| *mnp-1(ok2434)* | 1.929 | 0.876–3.086 | 0.274864209 | 0.001 | 0.113966678 |
| N2/*pL4440* | 19.488 | 14.868–26.847 | 2.939735872 | 0.5819 | 1.151364764 |
| N2/*pL4440-mnp-1* | 4.885 | 3.481–6.725 | 0.064467046 | 0.0025 | 0.28860924 |
| *mnp-1* rescue | 10.169 | 7.396–14.637 | 4.984458479 | 0.305 | 0.600791681 |
| *R03G8.6(ok3143)* | 7.52 | 5.915–9.634 | 0.751267595 | <0.0001 | 0.444286896 |
| *F19C6.4(ok2392)* | 4.822 | 2.786–8.285 | 0.715529874 | <0.0001 | 0.284887156 |
| *R03G8.6* rescue | 13.549 | 10.145–18.751 | 1.15912165 | 0.1493 | 0.800484462 |
| *F19C6.4* rescue | 13.971 | 10.166–20.133 | 2.386445544 | 0.3793 | 0.825416519 |

**App6Aa**

| Strain | LC$_{50}$ (μg/ml)[a] | 95% confidence interval (μg/ml) | Standard Deviation | *p* value relative to N2 | Relative sensitivity[b] |
|---|---|---|---|---|---|
| Wild type (N2) | 8.323 | 6.575–10.575 | 1.820689521 | / | |
| *mnp-1(ok2434)* | 7.661 | 4.385–13.525 | 0.253444932 | 0.739 | 0.920461372 |

**Cry21Aa**

| Strain | LC$_{50}$ (μg/ml)[a] | 95% confidence interval (μg/ml) | Standard Deviation | *p* value relative to N2 | Relative sensitivity[b] |
|---|---|---|---|---|---|
| Wild type (N2) | 1.336 | 1.053–1.774 | 0.139657915 | / | |
| *mnp-1(ok2434)* | 0.369 | 0.279–0.489 | 0.015524175 | 0.0003 | 0.276197605 |

**Cry5Ba**

| Strain | Median Survival (Days)[c] | 95% confidence interval | *p* value relative to N2 |
|---|---|---|---|
| Wild type (N2) | 3.533 | 2.877–4.189 | / |
| *mnp-1(ok2434)* | 1.333 | 0.889–1.777 | <0.001 |

**OP50**

| Strain | Median Survival (Days)[c] | 95% confidence interval | *p* value relative to N2 |
|---|---|---|---|
| Wild type (N2) | 11.633 | 9.693–13.574 | / |
| *mnp-1(ok2434)* | 10.655 | 8.614–12.696 | 0.585 |

[a] The LC$_{50}$ was determined by PROBIT analysis.

[b] The relative sensitivity was calculated by LC$_{50}$ of mutant/LC$_{50}$ of N2.

[c] The median survival was determined by Kaplan-Meier analysis.

To analyze whether the hypersensitivity of *mnp-1* mutant animals to Cry PFTs is not due to general sensitivity of these animals to different stresses, we tested the sensitivity of *mnp-1 (ok2434)* mutant to two other toxic chemical compounds, the heavy metal CuSO$_4$ or the oxidative stress agent H$_2$O$_2$ [34]. The dose-dependent mortality assays showed that *mnp-1(ok2434)* mutant animals have a similar sensitivity as wild type N2 worms when exposed to either CuSO$_4$ (S4A Fig) or H$_2$O$_2$ (S4B Fig). These data show that *mnp-1* mutant worms are specifically hypersensitive to certain nematicidal Cry PFTs.

To determine whether MNP-1 response involves the up regulation of *mnp-1* gene, we determined *mnp-1* gene expression by quantitative RT-PCR (qPCR) in N2 worms exposed to the nematicidal Cry PFTs, CuSO$_4$ or H$_2$O$_2$ as described in Materials and Methods. The expression of *mnp-1* gene in N2 worms exposed to *E. coli* food OP50 at the same condition was used as control. Fig 2E shows that expression of *mnp-1* gene was significant induced by exposure to Cry5Ba or Cry21Aa toxins, but showed no significant differences when worms were exposed to App6Aa, CuSO$_4$ or H$_2$O$_2$ compared to worms treated with the OP50 food control (Fig 2E). These data indicate that *mnp-1* gene was up regulated as a specific response to the exposure of *C. elegans* to the three domain nematicidal Cry5Ba and Cry21Aa PFTs.

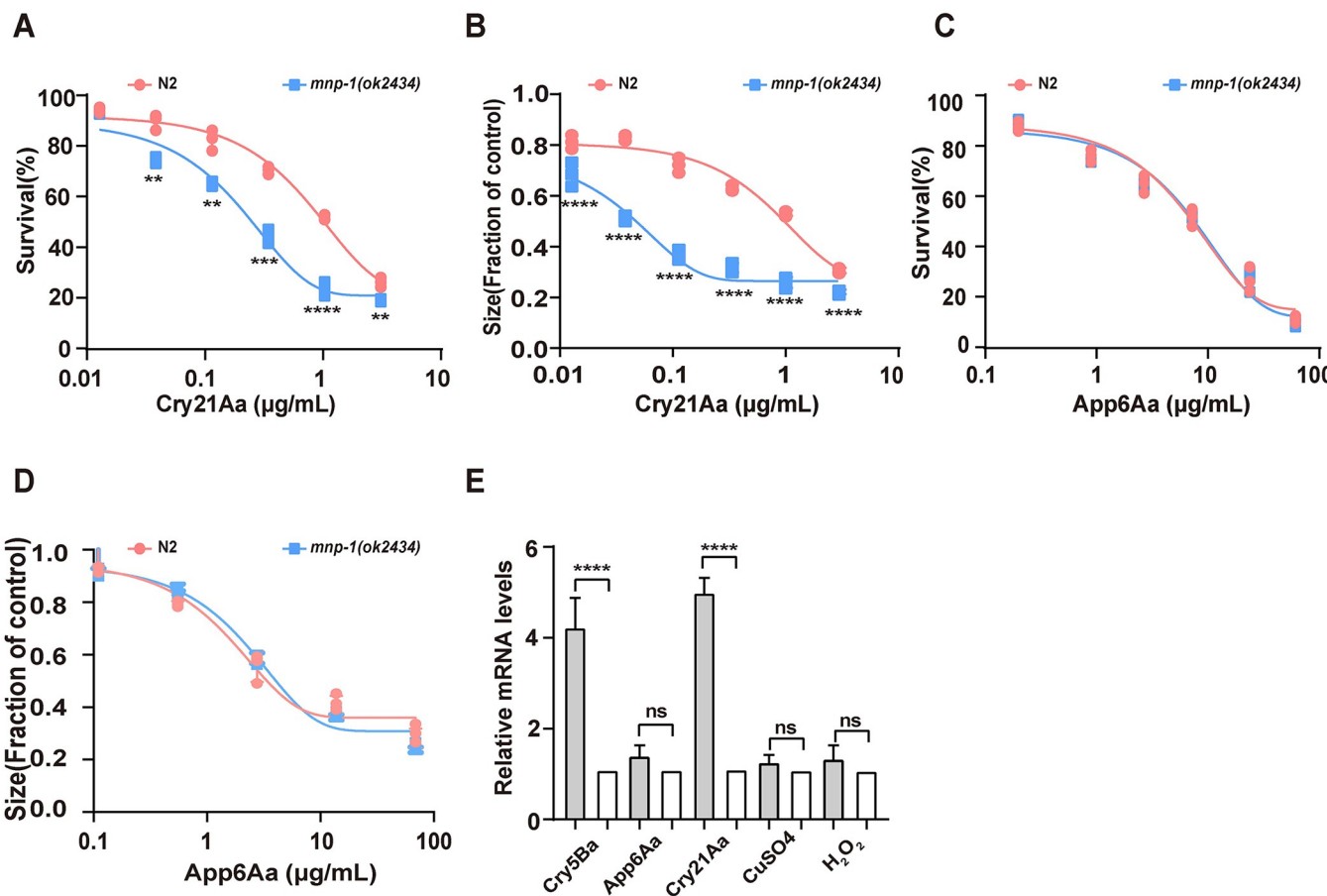

**Fig 2. MNP-1 is specifically involved in the defense of *C. elegans* to nematicidal Cry PFTs.** The dose-dependent mortality assays (A and C) and growth inhibition assays (B and D) comparing sensitivity of *mnp-1(ok2434)* mutant and wild-type N2 worms exposed to Cry21A (A and B), and App6Aa (C and D) toxins. Lethality was determined after 4 days of toxin exposure. $N = 3$ independent experiments contain three replication of at least 30 worms. (E), the relative expression of *mnp-1* gene in *C. elegans* upon exposure to nematicidal Cry toxins or to $CuSO_4$ and $H_2O_2$ chemicals determined by qPCR. The mean and SD values of three independent experiments are shown. Data points represent the mean values of three independent replicates, error bars denote the SD in (A–E). The *p*-value was determined by a Two-way ANOVA or Unpaired *t*-test (Welch's correction for unequal variances), ****$p < 0.00001$, ***$p < 0.001$, **$p < 0.01$, *$p < 0.05$ show significant differences and ns indicate no significant difference.

## 3. MNP-1 defense against Cry5Ba proceed through induction of proteases expression

The *mnp-1* gene encodes a membrane-associated APN that in *C. elegans* is required for embryonic muscle cell migration and neuronal cell migration [49, 50]. To analyze if MNP-1 response against PFTs involves the regulation of certain innate immunity pathways, we compared the genomic responses using microarrays of N2 and *mnp-1(ok2434)* mutant after treatment with Cry5Ba, with the aim to identify genes whose differential-regulation depends on MNP-1. The wild type N2 and *mnp-1(ok2434)* mutant animals were fed with *E. coli* cells expressing or non-expressing Cry5Ba toxin. Then their RNA was isolated, processed, and hybridized to Affymetrix arrays as described in Materials and Methods. The microarray analysis showed that 1575 genes from N2 have significant differential expression after Cry5Ba treatment (FC≥2.0, *p* ≥0.05) compared to the nematodes that were feed with *E. coli* cells without Cry protein expression. In the case of *mnp-1(ok2434)* mutant, we found that 755 genes showed significant differential expression after Cry5Ba treatment (FC≥2.0, *p* ≥0.05) compared with the control *E. coli* treatment (Fig 3A, detained information is shown in S1 Data). Gene ontology (GO) terms

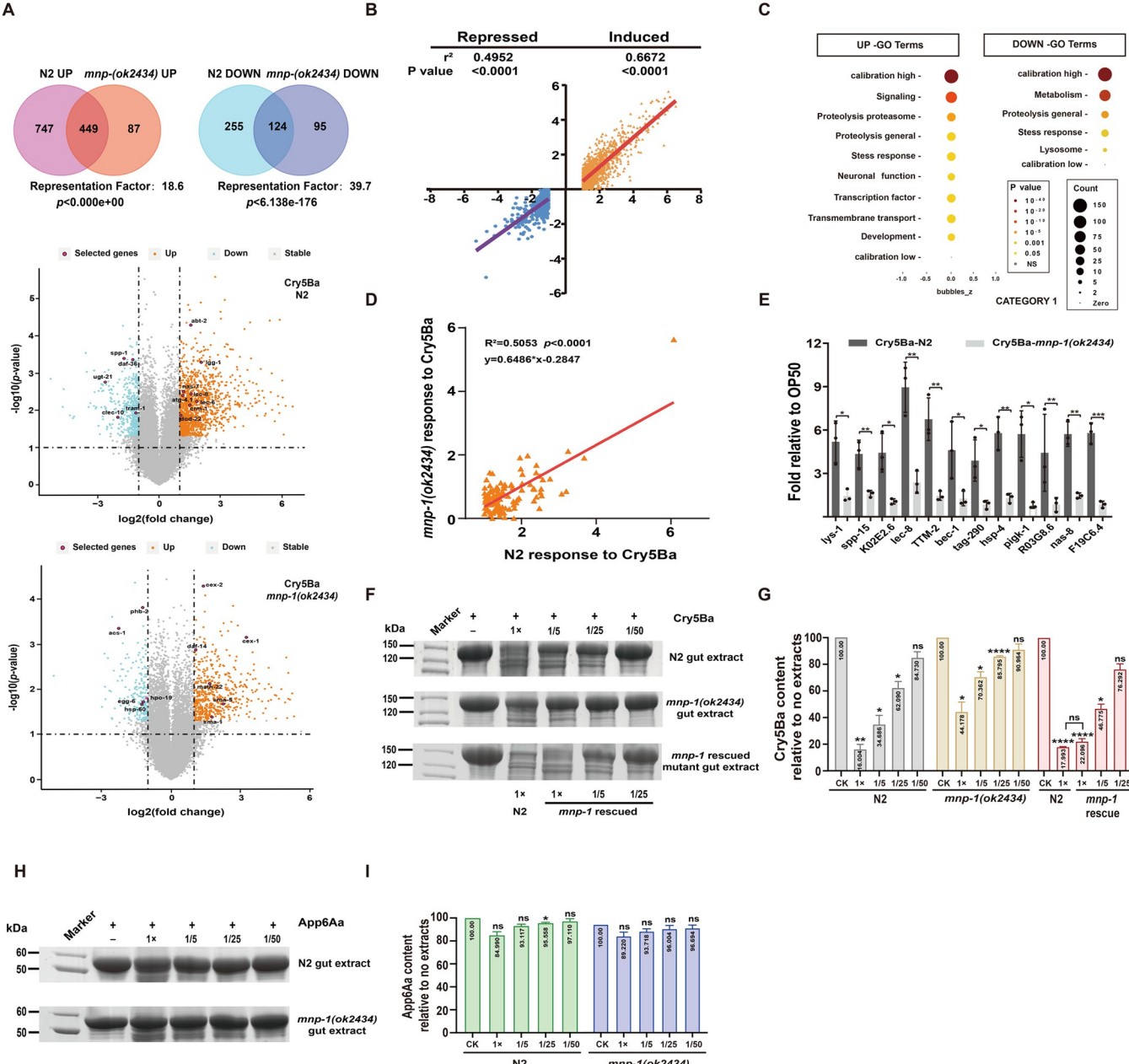

**Fig 3. MNP-1 defenses against Cry5Ba through the induction the peptidases that could mediated degradation of the toxin.** (A), Venn diagram and Volcano plot of differentially expressed genes (DEGs) in wild type N2 and *mnp-1(ok2434)* after Cry5Ba treatment. (B), the transcriptional responses to Cry5Ba in wildtype N2 versus *mnp-1(ok2434)* is plotted based on microarray data. Values are log2 scale transformations of the response to treatment. The subset of genes that were differentially regulated in N2 worms is shown ($p<0.05$ and fold change$>2$). Linear regressions are plotted for induced (orange) and repressed (blue) genes independently. In *mnp-1(ok2434)*, the induced and repressed genes response to Cry5Ba was largely intact ($r^2 = 0.6672$ and $0.4952$ respectively, $p<0.0001$ each). (C), gene ontology (GO) term analysis of up and down-regulated genes involved in Cry5Ba treatment affected by *mnp-1* mutation using Wormcat. Genes involved in signaling, proteolysis proteasome, proteolysis general, stress response and metabolism are enriched. (D), the up-regulated genes induced by Cry5Ba in *mnp-1(ok2434)* are dependent on *mnp-1* ($R^2 = 0.5053$, $p<0.0001$). (E), the relative transcription level of genes dependent by *mnp-1* upon Cry5Ba exposure was determined by qPCR. $N = 3$ independent experiments. (F), SDS-PAGE analysis of Cry5Ba degradation after incubation with crude extracts from different worms. Crude extracts were prepared form N2, *mnp-1(ok2434)*, or *mnp-1* recued worms, respectively. Then purified Cry5Ba sample was incubated with different dilution of these gut extracts, finally the presence of Cry5Ba in these samples was detected by SDS-PAGE analyses. The picture shows one of three independent experiments. (G), densitometry analysis of the images obtained with the different ratios of protein concentration to control after treatment is shown by the number within the column. $N = 3$ independent experiments. (H), SDS-PAGE analysis of App6Aa degradation with crude extracts at the same concentration from N2 and *mnp-1(ok2434)* mutant worms. A negative control without extract treatment is included in the figure. The picture shows one of three independent experiments. (I), densitometry analysis of the images obtained with the different ratios of protein concentration to control (App6Aa without gut extracts) after treatment is shown by the number within the column. $N = 3$ independent experiments. Data points represent the mean values of

three independent replicates, error bars denote the SD in (E, G and I). The *p*-value was determined by One-way ANOVA or Unpaired *t*-test (Welch's correction for unequal variances), ****$p < 0.00001$, ***$p < 0.001$, **$p < 0.01$, *$p < 0.05$ show significant differences and ns indicate no significant difference.

analysis of these microarray data showed that the differential gene expression in N2 and *mnp-1(ok2434)* mutant upon Cry5Ba treatment are mainly enriched in genes involved in defense responses, stress responses and metabolism (S5A and S5B Fig). To analyze if these differentially expressed genes are dependent on *mnp-1*, we compared the changes in expression of up and down regulated genes following Cry5Ba treatment between N2 and *mnp-1(ok2434)*. First, using the list of up and down regulated genes upon Cry5Ba exposure, we performed the separate linear regression analysis. The results revealed that both the induction and repression of genes in response to Cry5Ba were largely intact in *mnp-1(ok2434)* animals (r² = 0.6672 and 0.4952 respectively, *p*<0.0001 each) (Fig 3B). Then, analyzes of the microarray data, showed that 747 up regulated and 255 down regulated genes in Cry5Ba-treated N2 were affected by the *mnp-1* mutation (Fig 3A). Analysis of these data through WormCat, a *C. elegans* bioinformatics platform that facilitates the further refinement of functional categories within enriched groups, illustrated that signaling, proteolysis proteasome, general proteolysis and stress responses are the top categories within the UP class genes [51]. In the case of DOWN class genes, it was found that genes involved in metabolism, proteolysis and stress responses are the top three categories (Fig 3C and S3 Table). Since *mnp-1* mutant showed to be more sensitive to Cry5Ba toxicity, we proposed that it may be due to differences in genes involved in immune regulation. For this reason, we screened the 116 genes involved in proteolysis proteasome, general proteolysis and stress responses. Separate linear regression analysis showed that 106 genes were intact in *mnp-1(ok2434)* worms (r² = 0.5053, *p*<0.0001) (Fig 3D and S4 Table). In addition, we compared the average differential expression of these genes using a paired *t*-test. The average difference in induction ($X_\triangle = X_{mutant} - X_{wildtype}$) was calculated. This approach accounts for both the magnitude and direction of attenuation of the response to Cry5Ba, with negative values of $X_\triangle$ indicating attenuated induction. In *mnp-1(ok2434)* mutant, the average differences in induction of selected genes was significantly reduced compared with that of wild type (induction: $X_\triangle = -0.856$, *p*<0.0001). This analysis is consistent with the results of regression analysis (Fig 3D). Also, we compared the transcription levels of ten genes from the 116 genes involved in proteolysis proteasome, general proteolysis and stress responses, in N2 and *mnp-1(ok2434)* worms, to verify that *mnp-1* is involved in the induction of these genes upon Cry5Ba exposure. The transcription levels of the ten selected genes were lower in *mnp-1 (ok2434)* than in N2 (Fig 3E), which is consistent with the microarray data. All these results support that *mnp-1* is required for the induction of many genes involved in proteolysis proteasome, general proteolysis and stress responses during Cry5Ba intoxication.

To confirm that some genes regulated by *mnp-1* play a role in defending against Cry5Ba, we directed our attention to genes related in protein degradation. Among the genes found within the UP class, there are 27 genes involved in protein degradation (S4 Table). These data suggest that *mnp-1* may regulate the expression of certain proteases and thereby mediate the defense against Cry5Ba action. To analyze this hypothesis, crude extracts samples of N2 or *mnp-1* mutant worms were prepared as described in Materials and Methods and these samples were incubated with purified Cry5Ba protein to analyze their capacity for degradation of the Cry5Ba toxin. When Cry5Ba protein was incubated with 170 μg/ml crude extracts from N2 worms, 83.996% of this protein was degraded, while treatment of Cry5Ba with 34 μg/ml or 6.8 μg/ml crude extracts from N2 worms, resulted in 65.314% and 37.910% degradation of Cry5Ba protein, respectively. Fig 3F and 3G show the SDS-PAGE and statistical analysis of Cry5Ba band degradation after incubation with crude extracts at the same concentrations

from different worms. In the case of crude extracts from *mnp-1* mutant worms, treatment of Cry5Ba with 170, 34 and 6.8 μg/ml extracts, resulted in 55.822%, 29.638% and 14.205% degradation of Cry5Ba protein, respectively (Fig 3F and 3G). In agreement, crude extracts from *mnp-1* rescued worms showed recovered Cry5Ba degradation capability when compared with *mnp-1(ok2434)* mutant worms (Fig 3F and 3G), indicating the *mnp-1* gene was sufficient to restore the Cry5Ba degradation capability in the *mnp-1(ok2434)* mutant background. Finally, the crude extracts of N2 and *mnp-1* mutant worms showed no significant degradation of App6Aa protein (Fig 3H and 3I), supporting that MNP-1 is specifically involved in the defense against 3-domain Cry toxins.

## 4. Two proteases, F19C6.4 and R03G8.6, are responsible for MNP-1 mediated Cry5Ba degradation

To identify the proteases that are involved in Cry5Ba degradation, we screened the 19 proteases or protease inhibitor genes by gene silencing (RNAi) and analysis of Cry5Ba susceptibility of the silenced animals. S5 Table shows that silencing down two of these genes, *F19C6.4* and *R03G8.6*, resulted in a hypersensitive phenotype to Cry5Ba toxin intoxication similar to *mnp-1(ok2434)* mutant, suggesting that these proteases are involved in the defense response against Cry5Ba. The dose-dependent mortality assays confirmed that *F19C6.4* or *R03G8.6* silenced animals were significantly more sensitive to Cry5Ba compared with N2 worms (Fig 4A). The $LC_{50}$ values of Cry5Ba against *F19C6.4(ok2392)* and *R03G8.6(ok3143)* mutant worms were 4.822±0.716 μg/ml and 7.520±0.751 μg/ml, respectively, while the $LC_{50}$ value of Cry5Ba against N2 control was 16.926±0.8 μg/ml (Table 1), indicating 3.5- and 2.25-fold times higher susceptibility in the mutants than in control worms. Moreover, the rescued animals in *F19C6.4(ok2392)* and *R03G8.6 (ok3143)* mutant worms with the corresponding protease genes *F19C6.4* and *R03G8.6* expressed by their native promoters showed that the sensitivity to Cry5Ba of these rescued worms was comparable to that of wild type N2 worms, confirming that F19C6.4 and R03G8.6 are involved in Cry5Ba intoxication (Fig 4A). The qPCR analysis showed that the expression of *F19C6.4* and *R03G8.6* genes was induced after Cry5Ba toxin treatment, and that this expression depend on MNP-1 since *mnp-1(ok2434)* mutant did not induce the expression of these genes, showing no-significant difference in their expression in the presence or absence of Cry5Ba intoxication (Fig 4B). Furthermore, to determine if Cry5Ba induces the expression of F19C6.4 and R03G8.6 proteins we constructed transgenic worms expressing *F19C6.4p::gfp* and *R03G8.6p::gfp* fused proteins. Fig 4C show that F19C6.4 was expressed in the intestine, cephalic sheath cells, and coelomocytes, whereas R03G8.6 protein was primarily found in the nematode gonads (Fig 4C). The treatment of these transgenic nematodes with Cry5Ba toxin for 4 h showed a clear induction in the expression of F19C6.4 (Fig 4C and 4D). These data indicated that at least *F19C6.4* is expressed in the gut and is involved in MNP-1 mediated protection of *C. elegans* against Cry5Ba.

To further analyze if these two MNP-1 controlled protease proteins are responsible for Cry5Ba degradation, we prepared crude extracts from *F19C6.4(ok2392)* and *R03G8.6 (ok3143)* mutants, as well as their corresponding rescued worms, and analyzed their capacity to degrade Cry5Ba toxin. The Cry5Ba-degradation assays showed that the crude extracts from *F19C6.4 (ok2392)* or *R03G8.6(ok3143)* mutant worms had significantly lower degradation of Cry5Ba protein than the crude extracts from the wild type N2, in contrast to the rescued worms that showed a similar Cry5Ba-degradation capacity as the wild type N2 (Fig 4E and 4F). In addition, to verify if F19C6.4 and R03G8.6 proteins could degrade Cry5Ba *in vitro*, we cloned and expressed these two proteins in *E.coli* strain BL21(DE3) and these protease proteins were purified as indicated in Material and Methods. Both purified proteins were then incubated with purified Cry5Ba, Cry21Aa and App6Aa at 37°C. The results showed that both F19C6.4 and

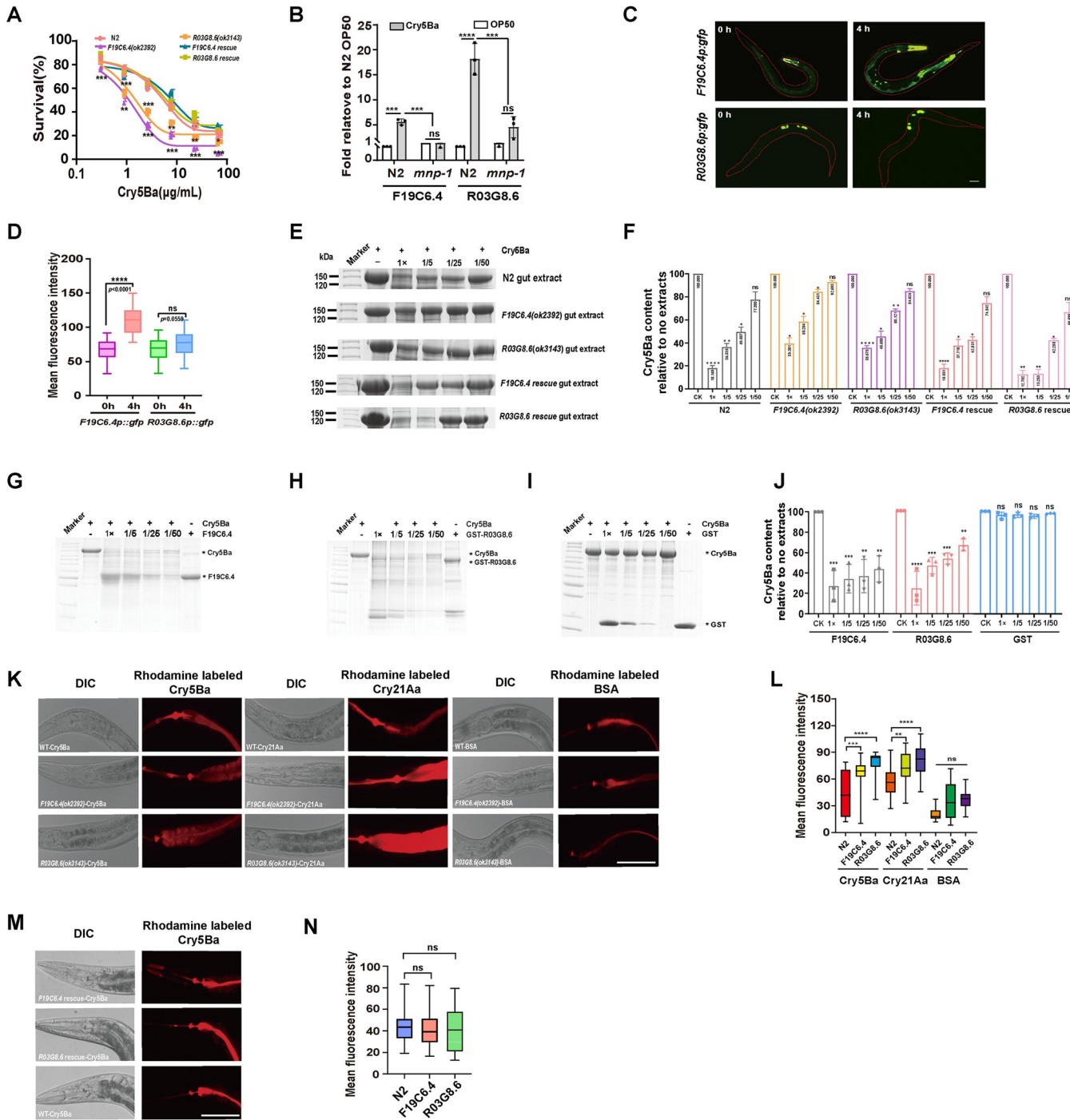

**Fig 4. Two proteases, F19C6.4 and R03G8.6, are responsible for MNP-1 mediated Cry5Ba degradation.** (A), a dose-dependent mortality assays comparing sensitivity of *F19C6.4(ok2392)* and *R03G8.6(ok3143)* mutants, *F19C6.4* and *R03G8.6* rescued strains with wild-type N2 worms exposed to purified Cry5Ba toxin. $N$ = 3 independent experiments, containing three replication of at least 30 worms. (B), analysis of the relative expression of *F19C6.4* and *R03G8.6* in N2 and *mnp-1(ok2434)* mutant worms upon exposure to Cry5Ba determined by qPCR. The mean and SD values of three independent experiments are shown. (C), fluorescence analysis of *R03G8.6p::gfp* and *F19C6.4p::gfp* worms after Cry5Ba treatment for 4 h. Transgenic worms were placed at plates containing BL21 expressing Cry5Ba for 4 h. Then imaged at 400 magnification on microscope. The bar denotes 100 μm. (D), quantification of mean fluorescence intensity of *R03G8.6p::gfp* and *F19C6.4p::gfp* worms after Cry5Ba treatment. $N$ = 3 independent experiments containing at least 20 worms. (E), SDS-PAGE analysis of Cry5Ba degradation with crude extracts from different worms. Crude extracts were prepared from N2, *mnp-1(ok2434)*, *F19C6.4(ok2392)* and *R03G8.6(ok3143)* mutants, F19C6.4 and R03G8.6 rescued worms, respectively. Then Cry5Ba protein was incubated with the different extracts for 1 h at the same concentration at 37°C, and analyzed by SDS-PAGE analyses. A negative control without extract treatment is included in the figure. (F), the ratio of Cry5Ba to the negative control after treatment with the gut extracts of N2, *mnp-1(ok2434)*, *F19C6.4(ok2391)* and *R03G8.6(ok3143)*. Numbers within the column in the figure are

showing the densitometry analysis of the bands. $N = 3$ independent experiments. (G-I), SDS-PAGE analysis of Cry5Ba with purified F19C6.4, GST-R03G8.6 and GST protein. Purified Cry5Ba was incubated with purified F19C6.4 (G), R03G8.6 (H) and GST(I) for 4 h at 37˚C, then the samples were analyzed by SDS-PAGE. The highest concentrations of F19C6.4, R03G8.6 and GST were 300 μg/ml. Different dilutions of F19C6.4, R03G8.6 and GST were used from 1* to 50*. A negative control without extract treatment is included in the figure. (J), the ratio of Cry5Ba to the negative control after treatment with the purified F19C6.4, R03G8.6 and GST. These results are the mean ± SD of three independent experiments. (K), rhodamine labeled-Cry5Ba is rapidly internalized into the midgut cells of F19C6.4 and R03G8.6 mutant worms. Wildtype and F19C6.4 and R03G8.6 mutant animals were fed rhodamine-labeled Cry5Ba or BSA for 40 min. Photographs were aquired with DIC and the rhodamine channel to visualize Cry5Ba, Cry21Aa or BSA. Bar denotes 100 μm. (L), quantification of pixel intensity of Cry5Ba or BSA in the intestinal cells. $N = 3$ independent experiments containing at least 20 worms. (M), analysis of rhodamine labeled-Cry5Ba gut cell internalized by wild type N2, F19C6.4 and R03G8.6 rescued worms. Photographs were aquired with DIC and the rhodamine channel to visualize the rhodamine labeled-Cry5Ba. Bar denotes 100 μm. (N), quantification of pixel intensity of Cry5Ba in the intestinal cells. $N = 3$ independent experiments containing at least 20 worms. Data points represent the mean values of three independent replicates, error bars denote the SD in (A, B, F, J and H). The *p*-value was determined by One-way ANOVA, Two-way ANOVA or Unpaired *t*-test (Welch's correction for unequal variances), ****$p < 0.00001$, ***$p < 0.001$, **$p < 0.01$, *$p < 0.05$ show significant differences and ns indicate no significant difference.

GST-R03G8.6 degraded Cry5Ba and Cry21Aa but not App6Aa (Figs 4G–4J and S6A–S6H). Furthermore, it was previously shown that rhodamine-labeled Cry5Ba could bind to the receptors, and after that it was internalized in the gut cells where it colocalized with granules in wild-type animals [29]. We hypothesized that if F19C6.4 and R03G8.6 proteases participate in the Cry5Ba and Cry21Aa degradation *in vivo*, this degradation process may attenuate the rate of toxins entering into the intestinal cells. Here, we fed rhodamine-labeled Cry5Ba or -labled Cry21Aa to L4-stage of *F19C6.4(ok2392)*, *R03G8.6(ok3143)* mutants, wild-type, and to *F19C6.4(ok2392)* and *R03G8.6(ok3143)* rescued worms for 40 min. Image analysis showed that rhodamine labeled-Cry5Ba and -Cry21Aa were rapidly internalized into gut cells of *F19C6.4(ok2392)* or *R03G8.6(ok3143)* mutant worms in contrast to that of wild-type and *F19C6.4* and *R03G8.6* rescued worms (Fig 4K–4N). Overall, these data confirmed that Cry5Ba treatment induce transcription of F19C6.4 and R03G8.6 proteases that is dependent in *mnp-1* and also that these proteases participate in Cry5Ba toxin degradation in *C. elegans*, although, other proteins may also participate in this defense response.

## 5. *MNP-1*-mediated *F19C6.4* and *R03G8.6* expression by activating FOXO transcription factor DAF-16

Both F19C6.4 and R03G8.6 belong to metalloproteinases, and their mutant nematodes exhibit reduced ability to degrade Cry5Ba (Fig 4E–4F). It has been shown that *F19C6.4* is a transcriptional target of the TGF-Dauer pathway in adults, while the transcription level of *R03G8.6* is significantly lower in *daf-16(mgDf50)*; *daf-2(e1370)* compared to *daf-2(e1370)*, suggesting that F19C6.4 and R03G8.6 may be influenced by the DAF-2/DAF-16 signaling pathway [52, 53]. The *C. elegans* DAF-2/DAF-16 signaling pathway has also been reported to participate in the defense of the nematode against external pathogens, with DAF-16 acting as a transcriptional regulator of many stress responses and antimicrobial genes [32]. We hypothesized that the transcription factor DAF-16 may be playing a role in MNP-1 mediated PFTs protection. To analyze this, we quantified the mRNA levels of *mnp-1*, *F19C6.4* and *R03G8.6* in wild type N2 worms and in *mnp-1(ok2434)* and *daf-16*(mu86) mutant worms fed with *E. coli* cells expressing or non-expressing Cry5Ba. Upregulation of *F19C6.4* and *R03G8.6* by Cry5Ba exposure was abolished in *mnp-1(ok2434)* and in *daf-16(mu86)* mutants (Fig 5A and 5B), indicating that *F19C6.4* and *R03G8.6* protease are regulated by both *mnp-1* and DAF-16. In addition, under normal conditions, *R03G8.6* expression was downregulated in *daf-16(mu86)* mutant, confirming that basal expression of *R03G8.6* is dependent on *daf-16* (Fig 5B). These data support that the DAF-16 activity is required for the induction of these proteases. We also tested whether mutations in DAF-16 protein affected Cry5Ba degradation. The crude extracts from N2, *mnp-1(ok2434)* and *daf-16(mu86)* mutant worms were prepared to analyze the degradation of Cry5Ba. Fig 5C and

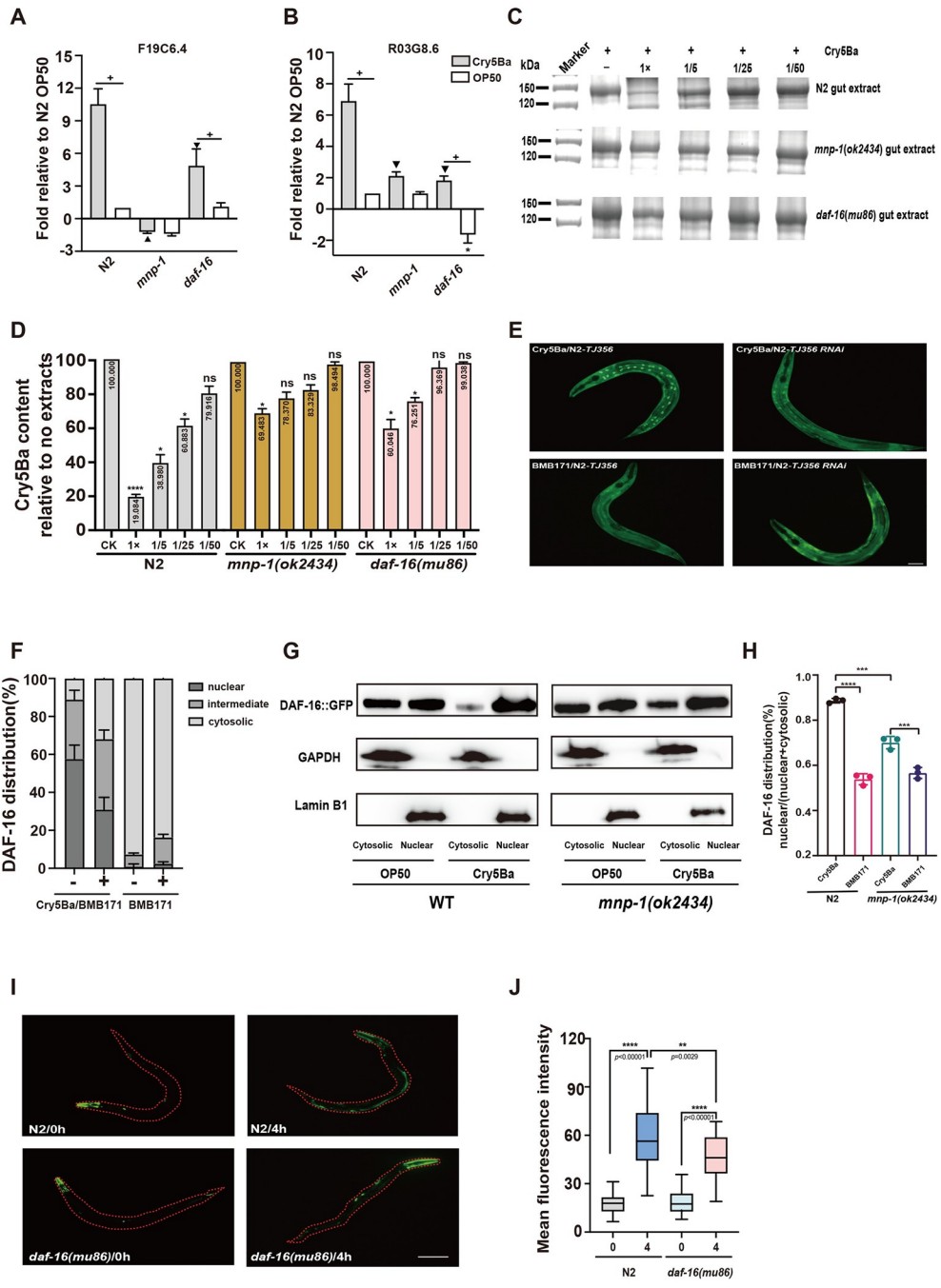

**Fig 5. *mnp-1* mediated the protease F19C6.4 and R03G8.6 expression by activating FOXO transcription factor *daf-16*.** (A-B), the relative expression level of *F19C6.4* (A) *and R03G8.6* (B) genes in N2, *mnp-1(ok2434)* and *daf-16* (mu86) mutant upon exposure to Cry5Ba and to control *E.coli* OP50 cells. The mean and SD values of three independent experiments are shown. Asterisks indicate student's *t*-test significant differences * *p*<0.05 comparison to N2 OP50, + *p*<0.05 comparing Cry5Ba to OP50, ▼ *p*<0.05 comparison to N2 Cry5Ba. (C), represntative images from SDS-PAGE analysis of Cry5Ba degradation with crude extracts from different worms. Crude extracts were prepared from N2, *mnp-1(ok2434)* and *daf-16*(mu86) mutant, respectively. Purified Cry5Ba was incubated with the different extracts at the same concentration for 1 h at 37°C, then the samples were analyzed by SDS-PAGE. A negative control without extract treatment is included in the figure. (D), the ratio of Cry5Ba to the negative control after treatment with the N2, *mnp-1(ok2434)* and *daf-16(mu86)* gut extracts. Numbers within the column in the figure are showing the densitometry analysis of the bands. *N* = 3 independent experiments. (E), representative fluorescence micrographs showing the localization of DAF-16::GFP expressed in adult worms. Nuclear localization of DAF-16::GFP fusion protein in N2-*TJ356(daf-16::gfp)* after exposure to BMB171/Cry5Ba. In contrast, no nuclear localization of DAF-16::

GFP fusion protein was observed in *mnp-1* silenced worms (named N2-*TJ356(daf-16::gfp-mnp-1* RNAi) by RNAi after exposure to BMB171/Cry5Ba, neither after exposure of both strains N2-*TJ356(daf-16::gfp)*, or N2-*TJ356(daf-16::gfp)-mnp-1*RNAi, to the control BMB171 strain, without toxin expression. Bar denotes 100 μm. (F), localization of DAF-16::GFP in N2-*TJ356(daf-16::gfp)* or N2-*TJ356(daf-16::gfp)-mnp1* RNAi. Categorized as showing predominately nuclear ("nuclear"), intermediate and cytosolic localized ("cytosolic") intestinal DAF-16::GFP upon exposed to Cry5Ba."+" silenced *mnp-1* N2-*TJ356(daf-16::gfp)- mnp1 RNAi*, "-"not silenced N2-*TJ356(daf-16::gfp)*. *N* = 3 independent experiments containing at least 30 worms. (G), the DAF-16 protein nuclear localization was reduced in *mnp-1(ok2434)* compared to WT N2 worms upon Cry5Ba exposure. Nuclear and cytosolic fractions were separated after the wildtype and *mnp-1*mutants were treated by Cry5Ba or OP50 for 4 h. Then proteins were analyzed by SDS-PAGE, and DAF-16 protein was detected by western blot using primary anti-DAF-16 antibody. Lamin B1 was used as a nuclear reference, and GAPDH was used as a cytosolic reference. The blot is one of the three independent experiments. (H), analysis of intensity of DAF-16::GFP in cytosolic and nuclear. The ratio of DAF-16::GFP to GAPDH or Lamin B1 represents the content of DAF::GFP in the cytosolic and nuclear respectively. *N* = 3 independent experiments. (I), *daf-16* mediated the protease F19C6.4 expression upon Cry5Ba treatment. Cry5Ba induced F19C6.4 expression was inhibited in *daf-16 (mu86)* that expressing *F19C6.4p::gfp*. Bar denotes 100 μm. (J), quantification of mean fluorescence intensity of *F19C6.4p::gfp*. *N* = 3 independent experiments containing at least 20 worms. Data points represent the mean values of three independent replicates, error bars denote the SD in (A, B, D, F, H and J). The *p*-value was determined by One-way ANOVA, Two-way ANOVA or Unpaired *t*-test, ****$p < 0.00001$, ***$p < 0.001$, **$p < 0.01$, *$p < 0.05$ show significant differences and ns indicate no significant difference.

5D show that the crude extracts from mutant worms were significant less efficient to degrade Cry5Ba than the extracts from wild type N2 control worms, supporting that DAF-16 is involved in Cry5Ba degradation through the induction of F19C6.4 and R03G8.6 proteases.

The *daf-16* gene codifies for a transcription factor that activates gene transcription when translocated into the nucleus. As described above, DAF-16 activity is required for the induction of the proteases. Therefore, we analyzed the effect of Cry5Ba exposure on the cellular localization of the DAF-16 using transgenic worms N2-*TJ356(daf-16::gfp)* that express a functional DAF-16::GFP fusion protein [54, 55]. As expected, we found that exposure to Cry5Ba caused significant increase of DAF-16 inside the nucleus compared with the control without toxin treatment (Fig 5E and 5F). To determine the possibility that *mnp-1* has a role in DAF-16 translocation into the nucleus after Cry5Ba treatment, we inactivated *mnp-1* by RNAi in N2-*TJ356(daf-16::gfp)* transgenic worms to analyze DAF-16 translocation into the nucleus when treated with Cry5Ba. We found that silencing *mnp-1* expression by RNAi significantly reduces the DAF-16::GFP localization into the nucleus after Cry5Ba treatment, compared to DAF-16::GFP localization in the transgenic worms N2-*TJ356(daf-16::gfp)* after Cry5Ba treatment (Fig 5E). Fig 5F shows the quantitative analysis of DAF-16::GFP localization in the different strains categorized as nuclear or cytosolic (named as intermediate and cytosolic) confirming that DAF-16::GFP localization into the nucleus increased after Cry5Ba treatment. In addition, we conducted western blot analyzes with cytosolic or nuclear samples purified from the worms to further confirm the localization of DAF-16 protein after Cry5Ba treatment. The results shows that Cry5Ba-treatment significantly induce DAF-16 protein localization into the nucleus sample, and this was reduced in *mnp-1(ok2434)* mutant worms (Fig 5G and 5H). Furthermore, to verify that DAF-16 nuclear localization activates F19C6.4 protease expression, we constructed the transgenic animals that express *F19C6.4p::gfp* in N2 and in *daf-16(mu86)* worms to compare the expression level of F19C6.4 in these worms under Cry5Ba treatment. The results showed that after 4 h treatment with Cry5Ba, the expression level of F19C6.4 is lower in *daf-16 (mu86)* than in N2 (Fig 5I and 5J). Overall, these data support that upon Cry5Ba exposure, *mnp-1* could induce DAF-16 nuclear localization to activate F19C6.4 protease expression.

## Discussion

APNs are endoproteases that could catalyze the cleavage the neutral amino acids from the N-terminus of protein [43]. In insects, APNs serve as Cry protein receptor, binding different Cry

proteins and participating in the pore-formation process [40, 41]. The typical lepidopteran APNs proteins possesses six identified motifs and two conserved domains known as M1_APN_1-like and ERAP1_C [56]. In *C. elegans*, *mnp-1* gene encodes a 781 amino acid APN protein that contains M1 peptidase domain and is required for embryonic muscle cell migration and neuronal cell migration [49, 50]. Conserved domain and motif analyze of MNP-1, showed that this protein lacks the conserved motifs of *Plutella xylostella* APNs. Moreover, MNP-1 is presumed to be catalytically inactive because it lacks three of the four essential zinc-binding amino acids in its HENNH + E motif (S1 Fig) [50, 56]. Based on the differences in conserved domains and motifs between MNP-1 and insect APNs, which serve as Cry protein receptors, it can be inferred that MNP-1 differs from insect APNs. Here, we showed that MNP-1 functions as a regulator of intestinal innate immunity effectors to protect *C. elegans* against PFTs. This is the first report that shows a that MNP-1 is associated with the host defense against PFT, supporting a new function for MNP-1 in *C. elegans*.

Cry5Ba exhibits high toxicity to parasitic and free-living *C. elegans* nematodes [36]. This toxin binds to the membrane receptor glycolipids and cadherin, and disrupt intestinal cells through forming pores on the cell membrane [31, 39]. Meanwhile, worms initiate different pathways to resist Cry5Ba intoxication. Membrane damage activates p38/MAPK that either induces TTM-2-dependent PFTs defense or IRE-1 dependent endoplasmic reticulum UPR to resist Cry5Ba toxin action. In addition, Rab-5 dependent endocytosis and Rab-11 dependent exocytosis machinery assist worms to eliminate the membrane-bound Cry5Ba. For the Cry5Ba that enters the cell, autophagy could be activated by transcription factor HLH-30/TFEB, the Cry5Ba encapsulated into autophagosomes finally fuses with lysosomes for further degradation [25, 35, 38, 57]. Here, we show that MNP-1 is involved in Cry5Ba defense in *C. elegans*. The Cry5Ba treatment induces the up regulation of *mnp-1*, which is located in the upstream of *daf-16* pathway, and participates in inducing DAF-16 nuclear localization where it can activate the expression of F19C6.4 and R03G8.6 proteases to degrade Cry5Ba in the intestine (Fig 6). MNP-1 dependent Cry5Ba degradation in the intestine represents a novel strategy for *C. elegans* to resist PFTs. Recent studies have confirmed that treatment with Cry1Ac induces a rise in insect hormones levels within *P. xylostella*, subsequently triggering the activation of the MAPK cascade. The activated MAPK cascade regulates the phosphorylation level of the nuclear receptor fushi tarazu factor 1(FTZ-F1), resulting in the upregulation of non-receptor gene expression and downregulation of the receptor gene expression. Decreased expression of the receptor gene confers the resistance to Cry1Ac in *P. xylostella* [20, 21]. These findings indicate that the organisms employ a shared defense mechanism to modulate gut gene expression upon detection of pore-forming toxins.

It is worth noting that the WormBase information indicates that the *F19C6.4* and *R03G8.6* genes are expressed in the nematode intestine, but the transgenic nematodes that we generated showed that *F19C6.4* was expressed in nematode intestines, cephalic sheath cells, and coelomocytes, while *R03G8.6* was primarily expressed in L4 nematode gonad (Fig 4C)[58, 59]. We report here that Cry5Ba treatment can significantly increase the transcription level of *F19C6.4* and *R03G8.6* genes, Cry5Ba can also induce an increase in fluorescence in *F19C6.4p::gfp* but not in *R03G8.6p::gfp*. This could be due to the low expression of R03G8.6 in *C. elegans*' intestine, as Cry5Ba has not been shown to increase its expression in the intestine. We also show that the degree of Cry5Ba degradation in F19C6.4 and R03G8.6 mutants is higher than that of the *mnp-1(ok2434)* mutant worms (Fig 4E and 4F), suggesting that these two proteases are functionally redundant or that there may be additional proteases involved in the degradation of Cry5Ba. In addition, the Cry5Ba induced up-regulation of *F19C6.4* and *R03G8.6* genes was significantly inhibited in both *daf-16* and *mnp-1(ok2434)* mutants (Fig 5A and 5B). However, in *daf-16* mutant, Cry5Ba treatment significantly increased the transcription of both genes,

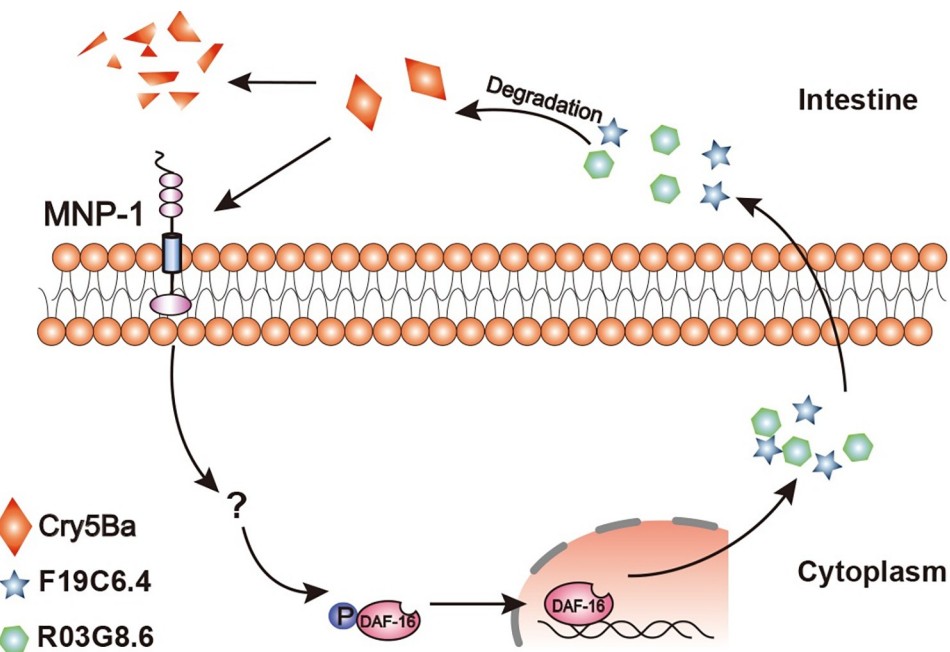

**Fig 6. Schematic diagram of Cry5Ba degradation mediated by MNP-1.** Cry5Ba treatment induces *mnp-1* up transcription which activates the DAF-16 nuclear localization. DAF-16 nuclear distribution induces expression of F19C6.4 and R03G8.6 proteases to degrade Cry5Ba in the intestine.

indicating that these two genes were also regulated by other pathways besides DAF-16. Furthermore, it remains to be determined the mechanism by which Cry5Ba or Cry21Aa trigger the MPN-1 response. Pore formation is unlikely to be involved, since App6Aa, a different PFT, did not trigger this response. An interesting possibility is that Cry5Ba or Cry21Aa trigger MPN-1 response by a direct interaction of these toxins with MPN-1 since this is a membrane protein. This remains to be analyzed.

Interestingly, insect midgut proteases such as chymotrypsin and trypsin activate the three-domain Cry protoxin at the N and C terminal regions to generate a 55–65 kDa activated toxin. Proteolytic processing of Cry protoxin in the gut lumen, is an important step, involved not only in the activation of Cry toxin fragment, but could also determine the specificity against insects [41]. Recently, the analysis of the midgut transcriptome response of the rice leaf-folder, *Cnaphalocrocis medinalis* (Guenée) after Cry1C toxin ingestion revealed that a large number of serine proteases, APNs, and carboxypeptidases were upregulated, implying that these proteases may be involved in the response of *C. medinalis* to Cry1C toxin action [60]. Similarly, analysis of *P. xylostella* midgut transcriptome response after intoxication with Cry1Ac toxin, revealed that two APN encoding genes are significantly up-regulated upon Cry1Ac treatment [61].Our work revealed a novel defense response for *C. elegans* against certain PFT in which MNP-1 activates DAF-16 nuclear distribution to modulate intestinal protease expression that degrade Cry5Ba protein. In addition, we also found that the protein extracts obtained from plant parasitic nematodes *Meloidogyne incognita* and *Ditylenchus destructor* could also degrade Cry5Ba (S7 Fig). Since the transcription factor DAF-16 and complex intestinal digestive enzymes are found from *C. elegans* to human, we propose that this novel host mechanism to eliminate PFTs via intestinal protease degradation could be widely used as a defense strategy for different animals including mammals to defend themselves against enteric bacterial pathogens. Thus, the knowledge gained through this study may help to develop novel strategies in the future to treat different diseases induced by pathogenic microorganisms.

## Materials and methods

### Bacterial strains, plasmids, and culture conditions

The bacterial strains and plasmids used in this study are listed in S6 Table. All *Escherichia coli* and Bt strains were grown on Luria-Bertani (LB) medium supplemented with the appropriate antibiotics at 37 or 28˚C, for *E. coli* or Bt, respectively.

### *Caenorhabditis elegans* strains and culture conditions

The wild-type *C. elegans* strain N2 and mutant strains were provided by the *Caenorhabditis* Genetics Center (CGC, http://www.cbs.umn.edu/CGC/) (S6 Table). Worms were cultured at 20˚C on NGM plates (0.3% NaCl, 0.25% tryptone and 1.5% agar) with *E. coli* strain OP50 as food source [62]. The synchronized L4 stage worms were prepared as previous described [63].

### Cry proteins preparation

Bt strains described in S6 Table were cultivated in liquid ICPM medium (0.6% tryptone, 0.5% glucose, 0.1% $CaCO_3$, 0.05% $MgSO_4$ and 0.05% $K_2HPO_4$, PH 7.0) at 28˚C, 220 rpm for three days until complete sporulation. The crystal inclusions were purified as previously described [64].

### RNA extraction and qRT-PCR assays

Total RNA was extracted from *C. elegans* using TRIzol reagent (Invitrogen, Carlsbad, California, USA). The cDNA was reverse transcribed with random primers using Superscript II reverse transcriptase (Invitrogen, Carlsbad, CA, USA) according to the manufacturer's protocol. The expression analyses of in *C. elegans* genes were performed using qPCR with the primers listed in S7 Table. The qPCR assays were conducted with Life Technologies ViiA 7 Real-Time PCR system (Life Technologies, California, USA) using the Power SYBR Green PCR Master Mix (Life Technologies, CA, USA) according to the manufacturers' instructions. The experiments were conducted in triplicate. Primer efficiency correction was done with $2^{-\triangle\triangle}Ct$ relative quantitation analyses using *tba-1* gene as reference and normalization.

### Microarray analysis

L1 stage synchronized N2 and *mnp-1 (ok2434)* worms were transferred to NGM plates spread with OP50 and incubated at 20˚C for about 44 h. The L4 stage worms were washed with M9 buffer for two or three times, and transferred to NGM plates spread with *E. coli* bacteria BL21 (pET28a) or BL21 (pET28a-*cry5B*) and incubated for additional 4 h at 20˚C. The worms were washed again with M9 buffer for two or three times, collected into 1.5 ml tubes, and stored at -70˚C after addition of 150 μl Trizol solution. Each treatment was performed three times. Microarray production, hybridization, and scanning assay were performed at Shanghai OE Biotech CO. Ltd. with Affymetrix *C. elegans* gene 1.0 ST. Default background subtraction and normalization settings were used. Spots were filtered based on foreground to background ratio; values less than 1.2 were flagged. Log (base 2) values were exported. Gene expression on BL21 (pET28a) or BL21 (pET28a-*cry5B*) were compared by *t*-test. Differences of 2 folds with *p*-values of 0.05 were considered significant. Sequencing reads were aligned to WormBase release WS235. Lists of upregulated genes used for comparisons were exported and further sanitized to remove dead genes and update Wormbase.ID to WormBase release WS270.

## Gene Ontology analyses

Genes that showed significant differential regulation were categorized as either upregulated (UP) or downregulated (DOWN). To analyze these groups for enriched gene classes based on Gene Ontology (GO) terms, publicly available online resources centered around *C. elegans* such as https://biit.cs.ut.ee/gprofiler/gost and http://wormcat.com/ were utilized. The Representation Factor was calculated at http://nemates.org/MA/progs/overlap_stats.html.

## RNA Interference (RNAi)

The RNAi assays were performed as previously described [44]. To induce dsRNA expression, *E.coli* strain HT115 transformed with the RNAi plasmids p*L4440-mnp-1*, and spread onto NG plates supplemented with 50 μg/ml ampicillin and 0.1mM isopropyl β-D-thiogalactopyrano-side (IPTG)), and incubated at 25°C overnight. *E.coli* HT115 with p*L4440-mnp-1* spread on the NG plates that did not contain the IPTG were used as control. The L1 stage worms were cultured on the dsRNA induction or non-induction plates until L4 stage. Then the L4 worms were washed out and transformed to the 96-well plates containing *E.coli* HT115 RNAi strain and purified Cry5Ba. After a further 5 days of incubation at 25°C, the survival rate of each well was recorded. Concentration of each toxin was analyzed in a triplicate set-up for each assay, and three independent assays were performed. The $LC_{50}$ values were determined by PROBIT statistical analysis [65].

## Nematode mortality assays

The Cry5Ba plate bioassays and $LC_{50}$ assays were conducted as previously described [66]. Plate assays were used to determine Cry5Ba's toxicity to the worms. Plates containing BL21 *E. coli* cells expressing Cry5Ba or an empty vector were prepared. We add 20 synchronized L4 worms per plate and incubated them at 20°C for 72 h. In addition, Cry5Ba liquid toxicity assay were used to quantify the $LC_{50}$ values of Cry5Ba against N2 and mutants. In brief, N2 and various mutant worms were exposed to purified Cry5B (the concentration range was 0.0–56 μg/ml) in S medium in 96-well plates with 20–30 worms per well to quantitatively score worms-survival after 5 days incubation at 25°C. Concentration of each toxin was analyzed in a triplicate set-up for each assay, and three independent assays were performed. The $LC_{50}$ values were determined by PROBIT statistical analysis [65].

## Nematode growth inhibition assays

L1 stage worms were fed for 48 h on control plates with OP50 food and different concentrations of Cry5Ba toxin. The relative health of each worm was evaluated qualitatively by comparing body size of N2 worms feed with OP50 food control.

## Life-span assays of *C. elegans*

A total of 20 L4 stage synchronized N2 and *mnp-1 (ok2434)* worms were transferred to fresh NGM plates spread with 300 μl ($OD_{600}$ = 0.3) *E. coli* strain OP50 used as food and containing 0.1 mg/ml FUDR to prevent eggs from hatching. The alive/dead worms were scored every 12 h. The worms that were alive were transferred to new NGM plates until all the worms were dead. Three independent biological repeats were performed.

## Construction of transgenic worms

The promoters of *F19C6.4*, *R03G8.6* and *mnp-1* were amplified by PCR using the primers: P-*F19C6.4*-F, 5'-GAAATAAGCTTGCATGCCTGCAGGGTTTCAAAATGGTTGGTA-3', P-

*F19C6.4*-R, 5'-GGGTCCTTTGGCCAATCCCGGGGATCCGTTTCATAAAATATCAGC-3',
P-*R03G8.6*-F, 5'- GAAATAAGCTTGCATGCCTGCAGGATACAAAAACTAACGA-3', P-*R03G8.6*-R, 5'- GGTCCTTTGGCCAATCCCGGGGATCCTTTTTGAAAAATCTACTG-3',
P-*mnp-1*-F, 5'- GAAATAAGCTTGCATGCCTGCAGGCCTGCAGGGTATGATGC
TTTTGGTG-3', P-*mnp-1*-R, 5'- GGTCCTTTGGCCAATCCCGGGGATCCGGATCCATGG
GAAGTAAGGAATA-3' (S7 Table). After that, the fragments were inserted into the pBS77
vector (a gift from Min Guo, Huazhong Agricultural University, contains the sequence of GFP
and 3'UTR(*Ce-unc-54*)) previously digested with SbfI and BamHI. The recombinant plasmids
(80 ng/μl) were injected to the gonads of wildtype N2, *daf-16(mu86)* or *mnp-1(ok2434)* mutant
as well as the marker vector *lin44p::gfp* at 20 ng/μl. To construct *mnp-1*, *F19C6.4* and *R03G8.6*
rescued worms, a full-length genes of 1488 bp, 2244 bp and 2346 bp of the *F19C6.4*, *R03G8.6*
and *mnp-1* cDNA were generated, then we generate the native promoters of these three genes
and their coding sequences by overlap PCR, using the primers: P-*F19C6.4*-F, 5'- GAAATAA
GCTTGCATGCCTGCAGGGTTTCAAAATGGTTGGTA-3', F-*F19C6.4*-R, 5'- GGGTCCT
TTGGCCAATCCCGGGGATCCTCAGATTCCATAAGTGAA-3'; P-*R03G8.6*-F, 5'- GAAA
TAAGCTTGCATGCCTGCAGGATACAAAAACTAACGA-3', F-*R03G8.6*-R, 5'-GGTCCT
TTGGCCAATCCCGGGGATCCTCAGAGATTGCATTTCAT-3'; P-*mnp-1*-F,5'-GAAATA
AGCTTGCATGCCTGCAGGCCTGCAGGGTATGATGCTTTTGGTG-3', F-*mnp-1*-R, 5'- G
GTCCTTTGGCCAATCCCGGGGATCCGGATCCTTACGCAACAGCTGCTAA-3'. These
amplicons were then cloned into the pBS77 vector previously digested with SbfI and BamHI.
These three recombinant plasmids (100 ng/μl) were injected to the gonads of *F19C6.4
(ok2392)*, *R03G8.6(ok3143)* and *mnp-1(ok2434)* mutant worms, as well as the marker vector
*lin44p::gfp* at 20 ng/μl, to generate three independent transgenic lines.

## Fluorescence analysis

The ImageJ program (NIH) was used to analyze the GFP or rhodamine fluorescence intensity.
Worms were selected by software and the fluorescence intensity (total pixel strength/area) of
each nematode was determined at the same conditions. Nematodes expressing *F19C6.4p::gfp*
(L4 stage) and *R03G8.6p::gfp* (L3 stage) were collected, these worms were placed either on con-
trol plates with *E. coli* that did not express Cry5B (pET28a vector control) or on plates pre-
pared with *E. coli* expressing Cry5B (pET28a-Cry5B), with 30 nematodes per plate. These
plates were incubated at 25°C for 4 h, then worms were photographed on 2% agarose pads
using fluorescence microscope and phase contrast microscope (Olympus BX51, Olympus,
Tokyo, Japan). Three plates were tested per assay, and all experiments were repeated indepen-
dently three or four times.

## DAF-16 nuclear localization assay

N2 worms expressing TJ356 (*daf-16::gfp*) and its variant where *mnp-1* was silenced by RNAi
were transferred to ENG plates spread with BMB171 and BMB171 (pBMB0215) bacteria, for
2–3 h incubation at 30°C. The worms were then washed once with M9 buffer, and transferred
to a pad covered with 2% agarose for collecting the fluorescence micrographs using fluores-
cence microscope and phase contrast microscope (Olympus BX51, Olympus, Tokyo, Japan).
Location of DAF-16::GFP in individual worms were classified as: nuclear (dot enrichment)
referring to immune response to Cry5Ba, cytosolic (diffusion) referring to none immune
response to Cry5Ba, and intermediate (half of dot enrichment). The proportion of worms in
each category is a metric for comparing the extent of nuclear localization of DAF-16 between
populations.

## Nuclear and cytosolic distribution of DAF-16 protein

L4 stage were prepared as previous described [63]. Nuclear and cytosolic fractions were separated as described by Singh *et al* [67]. Proteins were analyzed by SDS-PAGE, and DAF-16 proteins were detected by western blot assay. Lamin B1 and GAPDH were used as markers for the nuclear and cytosolic fractions, respectively. Antibodies against GAPDH, DAF-16, and Lamin B1 were obtained from Cell Signaling Technology (CST; USA) and HUABIO (China).

## Preparation of crude extracts from *C. elegans*

Nematode were grown for 4–5 days in 250 ml batch of liquid culture, then they were collected in M9 buffer as previously described [63]. The extraction procedure of worms' crude extract sample was done as reported by Zhang *et al* [68], with some modifications. Approximately 500 mg worms were washed with Tris-HCl (pH 8.0) buffer for three times to replace M9 buffer. Worms were then grinded with TGrinder (OSE-Y30, TIANGEN BIOTECH (BEIJING) CO., LTD.). The supernatant obtained after centrifugation at 13,800 *x*g for 20 min was defined as nematode crude extracts. Protein concentration was determined as described below.

## Gene cloning, heterologous expression, and purification of R03G8.6 and F19C6.4 in *E. coli* cells

Total RNA was extracted from *C. elegans* according to the manual (Promega, Madison, WI, USA). First strand cDNA was synthesized from total RNA according to the kit instructions (Takara, Tokyo, Japan). After removing the signal peptide, F19C6.4 (57 bp removed from the N-terminal) and R03G8.6 (51 bp removed from the N-terminal) sequences were amplified using the primer: *F19C6.4-pET28a*-F, 5'- CCGCGCGGCAGCCATATGAAGTCAATCGAC ATATCT-3', *F19C6.4-pET28a*-R, 5'-TCGACGGAGCTCGAATTCTCAGATTCCATAAGT GAA-3' and *R03G8.6-pGEX6P-1*-F, 5'-CGCGGATCCTCCGAATTAGATGCTCGG-3', *R03G8.6-pGEX6P-1*-R, 5'- CCGCTCGAGTCAGAGATTGCATTTCAT-3'. The amplicons were cloned into pET28a and pGEX6p-1, respectively. Then, recombinant plasmids were transformed into *E. coli* strain BL21 (DE3), positive transformants were selected in LB plates containing 50 μg/ml kanamycin or 100 μg/ml ampicillin and then sequenced to confirm the gene was cloned without mutations. To purify the recombinant F19C6.4 and R03G8.6 proteins, the transformants were grown in LB medium with either 50 μg/ml kanamycin or 100 μg/ml ampicillin at 37˚C with sharking until the optical density OD600 reached 1.0. Isopropyl β-D-thiogalactopyranoside (IPTG) was then added to the final concentration of 0.1 mM for additional 12 h at 16˚C. Cells and suspended particles were collected and homogenized in a buffer containing 20 mM Tris-HCl, 10 mM imidazole, 150 mM NaCl, pH 8.0. After disruption by a high-pressure homogenizer (JNBIO, Inc., China) and centrifugation at 12,000 *x*g at 4˚C for 40 min, the supernatant was loaded onto a column equipped with $Ni^{2+}$ affinity resin (Ni-NTA; CWBIO, Inc., China). The column was washed twice with 10 ml buffer containing 20 mM Tris-HCl, 200 mM imidazole, 150 mM NaCl, pH 8.0. Finally, the recombinant protein was eluted with buffer containing 20 mM Tris-HCl, and 500 mM imidazole, pH 8.0. The protein was further purified by using anion-exchange chromatography (Source 15Q, GE Healthcare) and size-exclusion chromatography (Superdex-200 Increase 10/300, GE Healthcare). For anion-exchange chromatography, both high and low salt buffer were used. The high salt buffer consists of 20mM Tris HCl, 1M NaCl, pH 8.0, while the low salt buffer contains 20mM Tris HCl, pH 8.0. The buffer employed for the size-exclusion chromatography contained 25 mM Tris-HCl, 150 mM NaCl, pH 8.0.

## Quantification of crude extracts from *C. elegans*

10 mg Coomassie brilliant blue G-250 was dissolved in 5 ml ethyl alcohol and 10 ml ortho-phosphoric acid (85% (w/v)), and then diluted with $ddH_2O$ to a final volume of 100 ml. The final liquid solution was defined as G-250 dye. A total of 100 mg bovine serum albumin (BSA) was dissolved and diluted with $ddH_2O$ to a final volume of 100 ml at 1 mg/ml BSA standard solution. Six different volumes of BSA standard solution (0, 20, 40, 60, 80 and 100 μl) were added to six 20 ml colorimetric tubes, respectively. Then, a final volume of 1 ml, was completed in each tube with $ddH_2O$ and 3 ml G-250 dye was added and mixed. Absorbance at 595 nm was determined after 2 min incubation to draw the standard curve of BSA. A total of 60 μl nematode extracts was added into another colorimetric tube, and the final volume was completed with $ddH_2O$ up to 1 ml. The absorbance at 595 nm of these tubes was determined after addition of 3 ml G-250 dye and mixing. The protein concentration was calculated according to the standard curve of BSA.

## Cry5Ba, Cry21Aa and App6Aa degradation analysis

The worms' extracts form N2, *mnp-1 (ok2434)*, *F19C6.4(ok2392)*, *R03G8.6(ok3143)*, *daf-16 (mu86)*, *D. destructor* and *M. incongnita* were diluted to 170 μg/ml, final concentration. Then, each sample was diluted up to 1/5, 1/25, and 1/50-fold. Cry5Ba, Cry21Aa or App6Aa proteins were used as substrates for incubation with the different worms' extracts at 37˚C for 1 h. After incubation, the samples were analyzed in 10% SDS-PAGE. To test if F19C6.4, R03G8.6 and GST degrade Cry5Ba, Cry21Aa and App6Aa, 300 μg/ml purified proteins were mixed with Cry5Ba and App6Aa, incubated with 37˚C for 4 h, and then the samples were analyzed by 10% SDS-PAGE.

## Data analysis

All experiments were performed a minimum of three times. The data were analyzed using Statistical Package for the Social Sciences (SPSS version 13.0 Chicago, IL, USA) or GraphPad Prism 8. Graphical representations of the lethal concentration assays were generated using nonlinear regression analysis. Lifespan data was analyzed with Kaplan-Meier survival curves. $LC_{50}$ values of Cry5Ba, App6Aa, Cry21Aa against worms were determined by PROBIT analysis. The survival of each mutant worms was compared to that of the wild-type in GraphPad Prism 8 with the two-tailed t test. For the data where three or more values were compared, one way ANOVA or two way ANOVA were analyzed by using the Tukey post test. More details were shown in S8 Table. Statistical significance indicated as follows: ns indicates not significant, * indicates $p < 0.05$, ** indicates $p < 0.01$, *** indicates $p < 0.001$, **** indicates $p < 0.0001$.

## Supporting information

**S1 Fig. Sequence analysis of APN homologous proteins in *C. elegans*.** (A), phylogenetic analysis of *C. elegans*, *Plutella xylostella*, *Bombyx mori*, *Danaus plexippus* and *Manduca sexta* APNs. The phylogenetic tree was performed in MEGA5.1 software. (B), the exon/intron structures of the *apn* genes. The exons and introns were presented by red boxes and black lines, respectively. Blue boxes are upstream and downstream sequences (C), the conserved domains analysis in APNs, each domain was presented in different colored boxes as indicated in the figure. (D), WebLogo plots highlight amino acid in the FPCFDEPAFKATFNI and GYYRVNYD motifs in MNP-1 proteins.
(TIF)

**S2 Fig. Identification of the *mnp-1* gene expression stage.** *mnp-1* is expressed in the egg to adult stage. The native promoter of *mnp-1* gene was generated and inserted into the pBS77, then the recombinant plasmid was injected to the gonads of *mnp-1(ok2434)* mutant. Transgenic worms and eggs were observed under a fluorescence microscope. The bar denotes 10 μm.
(TIF)

**S3 Fig. Knock down *mnp-1* expression in wild type N2 worms.** The L1 larvae of *C. elegans* wild-type strain (N2) were seeded on to NGM/IPTG plates with RNAi bacteria that expressed *mnp-1* dsRNA until they gown up to L4 larvae. Total RNA of RNAi silenced or control worms was extracted, and the relative expression level of the *mnp-1* mRNA in both samples was determined by qPCR. The mean and SD values of three independent experiments are shown.
(TIF)

**S4 Fig. Dose-dependent mortality assay of N2 and *mnp-1* mutant after treatment with $CuSO_4$ and $H_2O_2$.** Dose-dependent mortality assay comparing sensitivity of *mnp-1* mutant and wild-type N2 worms exposed to $CuSO_4$ and $H_2O_2$. $N = 3$ independent experiments, containing three replication of at least 30 worms. Data points represent the mean values of three independent replicates, error bars denote the SD in A and B.
(TIF)

**S5 Fig. Gene ontology analysis of differentially expressed genes.** Gene ontology enrichment analysis of genes regulated by Cry5Ba in wildtype N2 (A), and *mnp-1(ok2434)* (B) animals. Enrichment analysis of top 15 significant upregulated or downregulated processes were shown.
(TIF)

**S6 Fig. SDS-PAGE analysis with App6Aa and Cry21Aa with purified F19C6.4, GST-R03G8.6 and GST protein.** Samples of purified App6Aa or Cry21Aa were incubated with purified F19C6.4 (A and E), GST-R03G8.6 (B and F) and GST (C and G) for 4 h at 37˚C, then the samples were analyzed by SDS-PAGE. The highest concentrations of F19C6.4, R03G8.6 and GST were 300 μg/ml. A negative control without extract treatment is included in the figure. (D and H), the ratio of App6Aa and Cry21Aa to the negative control after treatment with the highest concentration of purified F19C6.4, R03G8.6 and GST. Data points represent the mean values of three independent replicates, error bars denote the SD in (D and H). The *p*-value was determined by One-way ANOVA, ***$p < 0.001$ and ns indicate no significant difference.
(TIF)

**S7 Fig. SDS-PAGE analysis of Cry5Ba degradation with crude extracts from *M. incognita* and *D. destructor*.** (A), crude extracts were prepared from *M. incognita* and *D. destructor*, respectively. Purified Cry5Ba was incubated with the different extracts at the same concentration for 1 h at 37˚C, then the samples were analyzed by SDS-PAGE. A negative control without extract treatment is included in the figure. (B), the ratio of Cry5Ba to the negative control after treatment with the *M. incognita* and *D. destructor* gut extracts. Numbers within the column in the figure are showing the densitometry analysis of the bands. $N = 3$ independent experiments. Data points represent the mean values of three independent replicates, error bars denote the SD. The *p*-value was determined by Unpaired *t*-test, ****$p < 0.00001$, ***$p < 0.001$, **$p < 0.01$, *$p < 0.05$ show significant differences and ns indicate no significant difference.
(TIF)

**S1 Data. Excel spreadsheet containing, in separate sheets, the underlying numerical data for Figure panels 1A, 1B, 1C, 1E, 1F, 2A, 2B, 2C, 2D, 2E, 3E, 3G, 3I, 4A, 4B, 4D, 4E, 4J, 4L, 4N, 5A, 5B, 5D, 5F, 5H and 5J.**
(XLSX)

**S1 Table. Information of APNs proteins containing M1_APN_2 and ERAP1-C domains that were found in *C. elegans*.**
(XLSX)

**S2 Table. Toxicity data of Cry5Ba toxin against APN mutant and APN silenced worms.**
(XLSX)

**S3 Table. List of differentially expressed genes after Cry5Ba exposure in wildtype N2 but not in *mnp-1(ok2434)*.**
(XLSX)

**S4 Table. List of the differentially expressed genes involved in stress responses, proteolysis general and proteolysis proteasome that dependent on *mnp-1* after Cry5Ba treatment.**
(XLSX)

**S5 Table. RNAi clones that render the worms hypersensitive to Cry5Ba.**
(XLSX)

**S6 Table. Bacterial and *Caenorhabditis elegans* strains used in this study.**
(XLSX)

**S7 Table. Primers used in this study.**
(XLSX)

**S8 Table. Statistical analysis and *p* values of each figure.**
(XLSX)

## Acknowledgments

We thank the Caenorhabditis Genetics Center for worm strains. We are grateful to the associate professor Min Guo from Huazhong Agricultural University for the construction of transgenic animals.

## Author Contributions

**Conceptualization:** Feng Chen, Donghai Peng.

**Data curation:** Feng Chen, Cuiyun Pang, Wei Zhou, Danyang Xiao.

**Formal analysis:** Feng Chen, Ziqiang Zheng.

**Funding acquisition:** Ming Sun, Donghai Peng.

**Investigation:** Feng Chen, Cuiyun Pang, Wei Zhou, Zhiqing Guo, Danyang Xiao, Hongwen Du.

**Supervision:** Ming Sun, Donghai Peng.

**Visualization:** Feng Chen, Ziqiang Zheng.

**Writing – original draft:** Feng Chen, Cuiyun Pang.

**Writing – review & editing:** Feng Chen, Alejandra Bravo, Mario Soberón, Ming Sun, Donghai Peng.

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
