## [Decision Letter · Decision Letter 0]

7 Jun 2023

Dear Dr. Peng,

Thank you very much for submitting your manuscript "Aminopeptidase MNP-1 triggers intestine protease production by activating daf-16 nuclear location to degrade pore-forming toxins in Caenorhabditis elegans" for consideration at PLOS Pathogens. As with all papers reviewed by the journal, your manuscript was reviewed by members of the editorial board and by independent reviewers. The reviewers appreciated the attention to an important topic. Based on the reviews, we are likely to accept this manuscript for publication, providing that you modify the manuscript according to the review recommendations.

Sincerely,

Michael Wessels

Section Editor

PLOS Pathogens

Kasturi Haldar

Editor-in-Chief

PLOS Pathogens

orcid.org/0000-0001-5065-158X

Michael Malim

Editor-in-Chief

PLOS Pathogens

orcid.org/0000-0002-7699-2064

Reviewer Comments (if any, and for reference):

Reviewer's Responses to Questions

**Part I - Summary**

Reviewer #1: The authors have done a nice job responding to and addressing the criticisms I raised in my initial review of the manuscript.

Reviewer #2: This manuscript describes a new defense strategy against pore-forming toxin Cry5Ba in Caenorhabditis elegans. The authors revealed that MNP-1 up regulates two protease genes F19C6.4 and R03G8.6 through FOXO transcription factor DAF-16, which in turn leads to the degradation of Cry toxins. Overall, this is an important study and it will be of interest to scientists studying interactions between pore-forming toxins and their pathogenic hosts.

**Part II – Major Issues: Key Experiments Required for Acceptance**

Reviewer #1: (No Response)

Reviewer #2: (No Response)

**Part III – Minor Issues: Editorial and Data Presentation Modifications**

Reviewer #1: (No Response)

Reviewer #2: I think the current version is better than before, and I have some minor comments to help further improve it:

Minor comments:

Lines 257 and 280: The number of digits after the decimal point should be unified.

Lines 272-273: How to pick out “19 proteases or protease inhibitor genes” in “27 genes involved in protein degradation (Table S4)” for RNAi?

Line 294: R03G8.6 protein was primarily found in the nematode gonads, and the main target tissue of Cry toxins is gut. How to explain the role of R03G8.6 in degradation of Cry toxins?

Lines 437-439: Other defense strategies depending on APN against pore-forming toxin in insect need in-depth discussion (see Guo et al., PLoS Pathogens, 2021, 17: e1009917 and Guo et al., Nature Communications, 2022, 13: 6024).

Fig. 3 and Fig. 4: Some figures have low resolution and many letters in figures can’t be distinguished, please offer figures with high resolution; Fig. 4F is not fully displayed and the last histogram is hided.

PLOS authors have the option to publish the peer review history of their article (what does this mean?). If published, this will include your full peer review and any attached files.

Reviewer #1: No

Reviewer #2: **Yes: **Zhaojiang Guo

Figure Files:

Data Requirements:

Reproducibility:

References:

---

## [Editor Report · Decision Letter 1]

23 Jun 2023

Dear Dr. Peng,

We are pleased to inform you that your manuscript 'Aminopeptidase MNP-1 triggers intestine protease production by activating daf-16 nuclear location to degrade pore-forming toxins in Caenorhabditis elegans' has been provisionally accepted for publication in PLOS Pathogens.

In addition, we noticed a typographical error in the revised manuscript: in line 78, the word "statues" should be "status."  Please make this change when you prepare the final manuscript.

Best regards,

Michael Wessels

Section Editor

PLOS Pathogens

Kasturi Haldar

Editor-in-Chief

PLOS Pathogens

orcid.org/0000-0001-5065-158X

Michael Malim

Editor-in-Chief

PLOS Pathogens

orcid.org/0000-0002-7699-2064
---

## [Editor Report · Acceptance letter]

10 Jul 2023

Dear Dr. Peng,

We are delighted to inform you that your manuscript, "Aminopeptidase MNP-1 triggers intestine protease production by activating daf-16 nuclear location to degrade pore-forming toxins in Caenorhabditis elegans," has been formally accepted for publication in PLOS Pathogens.

Best regards,

Kasturi Haldar

Editor-in-Chief

PLOS Pathogens

orcid.org/0000-0001-5065-158X

Michael Malim

Editor-in-Chief

PLOS Pathogens

orcid.org/0000-0002-7699-2064